# HIV false positive screening serology due to sample contamination reduced by a dedicated sample and platform in a high prevalence environment

Michael A. Linström[1,2]*, Wolfgang Preiser[1,2], Nokwazi N. Nkosi[1,2], Helena W. Vreede[3,4], Stephen N. J. Korsman[3,5], Annalise E. Zemlin[1,6], Gert U. van Zyl[1,2]

1 National Health Laboratory Service, Tygerberg Hospital, Cape Town, South Africa, 2 Division of Medical Virology, Department of Pathology, University of Stellenbosch, Cape Town, South Africa, 3 National Health Laboratory Service, Groote Schuur Hospital, Cape Town, South Africa, 4 Division of Chemical Pathology, Department of Pathology, University of Cape Town, Cape Town, South Africa, 5 Division of Medical Virology, Department of Pathology, University of Cape Town, Cape Town, South Africa, 6 Division of Chemical Pathology, Department of Pathology, University of Stellenbosch, Cape Town, South Africa

* michael.linstrom@nhls.ac.za

**Data Availability Statement:** All relevant data are within the paper and its Supporting Information files.

## Abstract

Automated testing of HIV serology on clinical chemistry analysers has become common. High sample throughput, high HIV prevalence and instrument design could all contribute to sample cross-contamination by microscopic droplet carry-over from seropositive samples to seronegative samples resulting in false positive low-reactive results. Following installation of an automated shared platform at our public health laboratory, we noted an increase in low reactive and false positive results. Subsequently, we investigated HIV serology screening test results for a period of 21 months. Of 485 initially low positive or equivocal samples 411 (85%) tested negative when retested using an independently collected sample. As creatinine is commonly requested with HIV screening, we used it as a proxy for concomitant clinical chemistry testing, indicating that a sample had likely been tested on a shared high-throughput instrument. The contamination risk was stratified between samples passing the clinical chemistry module first versus samples bypassing it. The odds ratio for a false positive HIV serology result was 4.1 (95% CI: 1.69–9.97) when creatinine level was determined first, versus not, on the same sample, suggesting contamination on the chemistry analyser. We subsequently issued a notice to obtain dedicated samples for HIV serology and added a suffix to the specimen identifier which restricted testing to a dedicated instrument. Low positive and false positive rates were determined before and after these interventions. Based on measured rates in low positive samples we estimate that before the intervention, of 44 117 HIV screening serology samples, 753 (1.71%) were false positive, declining to 48 of 7 072 samples (0.68%) post-intervention (p<0.01). Our findings showed that automated high throughput shared diagnostic platforms are at risk of generating false-positive HIV test results, due to sample contamination and that measures are required to address this. Restricting HIV serology samples to a dedicated platform resolved this problem.

**Funding:** The author(s) received no specific funding for this work.

**Competing interests:** The authors have declared that no competing interests exist.

## Introduction

Laboratory-based diagnosis of Human Immunodeficiency Virus (HIV) infection in adults relies on 4[th] generation serological testing. In high-prevalence settings like South Africa, this requires a single screening test followed by one or two confirmatory tests for samples with reactive screening results [1]. Although screening is mostly done by point of care testing using rapid lateral flow chromatographic assays, samples with discordant rapid test results or from hospitalised patients are referred for laboratory-based testing. Due to high testing volumes, the public sector Tygerberg Hospital (TBH) National Health Laboratory Service (NHLS) laboratory has implemented automated 4[th] generation HIV serology on a high-throughput platform. Platforms that allow the analysis of different diagnostic tests across disciplines are often preferred due to the perceived economy of scale and efficient utilisation of space and personnel. These solutions are nevertheless not without challenges.

In a high prevalence setting, prior research by Hardie et al. (2017) found that repeatedly processing known HIV negative samples through automated platforms increased the probability of the sample subsequently testing positive [2]. A similar platform and setting exist at TBH NHLS where the Roche® Modular Pre-Analytic System (MPA) is used to process samples received for Clinical Chemistry, Immunology and Virology. Once samples are loaded, processing commences automatically, which entails centrifugation of the sample, sample aspiration into aliquots, directing samples to the appropriate testing module, and eventually sample storage.

TBH NHLS utilises an HIV serology screening assay which claims a 100% sensitivity and a 99.81% specificity for routine diagnostic samples [3]. The assay is said to detect p24 antigen on average five days after detectable viraemia and low-level HIV antibodies early after seroconversion, making it valuable for the early diagnosis of HIV in high incidence settings [4]. The high analytical sensitivity however poses a challenge for interpreting low positive (LP) results that may occur due to cross contamination of negative samples. Considering the high concentration of HIV antibodies and antigen in some patients, carry-over of microscopically small droplets may be sufficient to cause false positive (FP) results as HIV-positive patient samples remain positive on this platform even if diluted up to $10^{-5}$ [5]. Such FP results are determined by negative retesting of an independently obtained subsequent sample from a patient. Prompted by published evidence and investigation of individual cases that were FP, we suspected that sample contamination associated with pre-analytical processing on the MPA and testing on a high-throughput clinical chemistry module may be responsible for a large proportion of FP results.

### Our study aims were

1. To assess the factors that were associated with low positive and equivocal (LP&E) and FP HIV serology testing.

2. To investigate the impact of the dedicated sample on the overall prevalence of LP&E and FP results.

## Methods

Ethics approval was obtained from the Stellenbosch University's Health Research Ethics Committee (HREC) # N19/04/044. The data was fully anonymised and de-identified before analysis. The HREC waived the requirement for informed consent on the grounds of the research involved retrospective electronic laboratory test data and records.

## Study population

TBH NHLS Virology processes samples directly received from clinics and hospitals and referred from other laboratories in the drainage area which includes parts of the Cape Town Metropole, the Western Cape and the Northern Cape provinces.

Samples included were those sent to TBH NHLS for HIV screening serology (HIV-S) between 1 September 2017 (when the MPA was installed) and 31 May 2019. As creatinine (CRT) testing is a common clinical chemistry test it was used as proxy for a sample that had concurrent clinical chemistry testing and was therefore at risk of pre-analytical or clinical chemistry module-associated carry-over contamination.

## Laboratory tests

The Roche® Cobas® 6000 E601 analyser with the Roche® Elecsys® HIV combi PT fourth generation assay was used as the screening and the Abbott® ARCHITECT® HIV Ag/Ab Combo® as the confirmatory test for HIV serology. Both are 4th generation HIV serology assays, combining antibody with antigen detection. Both tests are sandwich microparticle immunoassays, where Abbott® uses chemiluminescence and Roche® uses electrochemiluminescence.

The Roche® Cobas AmpliPrep®/Cobas TaqMan® HIV-1 Qualitative Test version 2 (qualitative total nucleic acid PCR) and the Roche® Cobas AmpliPrep®/Cobas TaqMan® HIV-1 Quantitative Test version 2 (quantitative RNA PCR) were used as tie-breaker assays to determine infection status for patients with indeterminate HIV serology using a subsequent fresh new sample. These assays employ the principle of reverse transcriptase real-time polymerase chain reaction (PCR) to detect HIV nucleic acids.

## Dedicated sample implementation

Subsequent to our realization that samples may become cross-contaminated on the high throughput chemistry module we attempted to assure that samples for HIV-1 serology are tested on the virology module only. This intervention included a notice to clinicians to send a separate blood sample for HIV serology and once received in the laboratory appending a "Q" suffix to all HIV-1 serology sample labels, which prevents these samples from being recognized and tested on any other module than the dedicated virology module. We were able to extract data and compare rates before and after this intervention.

## Data retrieval and analysis

As molecular testing is often required in the work-up of discordant serology, the proportion of LP&E results that had a fresh independently collected sample for subsequent serological or molecular HIV test was assessed. The combined outcome of the subsequent independently collected sample's test results was then classified as negative, positive, and indeterminate. The median time between the original and subsequent fresh independently collected test result was used to indicate clinician response to LP&E results.

When a subsequent independently collected serology test or molecular test gave a negative result, initial results were classified as FP. When a subsequent independently collected follow-up serology result was high positive and confirmed on the confirmatory assay or the molecular test result was positive, the sample was classified as true positive. If subsequent testing produced an LP&E result like the original, with no additional molecular or serology testing results, samples were classified as indeterminate.

The FP rate was defined as the percentage of LP&E samples with a negative definitive result, as clinicians are required to send independently collected follow-up samples if an LP&E result is generated. It is expected that not all LP&E results have testing of subsequent independently collected follow-up samples due to the reliance on clinicians to provide such samples. Therefore, the false positive rate was extrapolated based on the LP&E results with known subsequent independently collected follow sample testing results. This rate was then used to calculate the prevalence of true positives, the true positive predictive value (PPV) and the specificity. The PPV calculated from the prevalence and the manufacturer's claimed specificity was used as comparison.

The outcome of confirmatory testing was described as overall proportion of FP results that generated a discordant confirmatory test result on a separate platform, bearing in mind that the sample could have been contaminated before confirmatory testing.

Results for HIV serology including HIV-S and HIV confirmatory (HIV-C) testing and subsequent tests linked to the same individual were retrieved from the NHLS TrakCare laboratory information system. As a single sample identifier (ID) is generated when different sample tubes linked to the same request form are registered, the same sample-ID for different tests indicates concurrent testing. However, it does not necessarily indicate testing of the same tube on a different instrument. Therefore, in order to identify samples at risk of contamination, we first used CRT registered on the same sample ID of HIV-S samples as a screen for being at risk of clinical chemistry module processing and contamination. Second, we used samples being tested on the clinical chemistry module first, rather than second to further stratify contamination risk. We could be confident that samples with CRT results released after HIV-S results did not pass through the clinical chemistry module first and would not have been a source of contamination risk.

LP&E and FP results were stratified into samples with concurrent CRT testing vs. those without, CRT results released first (CRT1st) vs. CRT results released second (CRT2nd), and before and after the dedicated sample intervention. These comparisons included prevalences and the calculation of odds ratios (OR) to determine the risk of generating a LP&E and FP result within these strata. Fischer's Exact Test was used to assess whether there were significant associations with LP&E or FP results, respectively before and after dedicated sample intervention, whereas the Mantel-Haenszel test was used to assess whether these associations were independent of the dedicated sample intervention.

## Results

A total of 51189 results of samples sent for HIV-S were included over the 21 month period as seen in Table 1. Of these, 41156 (80.4%) were resulted as negative, 9085 (17.7%) as positive and 946 (1.8%) as LP&E. Table 1 separates these values into totals found before and after the dedicated sample intervention and illustrates the proportional changes seen.

A summary of LP&E and FP results stratified by concurrent CRT and CRT1st vs CRT2nd is provided in Table 1. The largest proportion of LP&E results were in the equivocal range: 553 (59%) of the 946 overall non-stratified LP&E results (LP&E Total). This was not significantly different before and after the dedicated sample intervention. Of the 946 LP&E results, 485 (51%) had an independently collected follow-up sample for serological or molecular testing (Table 1). Of these 485 subsequent independently collected follow-up sample results, 411 (85%) were negative (making the initial result FP), 31 (6%) were positive (making the initial result true positive), and 43 (9%) remained indeterminate. 179 of the 485 independently collected follow-up samples (37%) had molecular testing performed on the follow-up sample instead of serological testing. The subsequent independently collected follow-up sample was

**Table 1. Summary of Low Positive and Equivocal (LP&E) HIV antibody rates in samples tested before and after introduction of a dedicated sample tube for HIV serology.**

| | | Before dedicated sample implementation | | After dedicated sample implementation | | |
|---|---|---|---|---|---|---|
| | | Number | Proportion | Number | Proportion | p-Value |
| **Original Sample** | **HIV-S Total** | 44117 | | 7072 | | |
| | **Negative** | 35141 | 79.66% | 6015 | 85.05% | NA |
| | **Positive** | 8091 | 18.34% | 994 | 14.06% | NA |
| | **LP&E Total** | 883 | 2.00% | 63 | 0.89% | <0.001 |
| | **HIV-S Only Total** | 26986 | | 3481 | | |
| | **HIV-S Only LP&E** | 354 | 1.31% | 42 | 1.21% | 0.602 |
| | **HIV-S/CRT Total** | 17130 | | 3591 | | |
| | **HIV-S/CRT LP&E** | 529 | 3.09% | 21 | 0.58% | <0.001 |
| | **CRT1st Total** | 15505 | | 2563 | | |
| | **CRT1st LP&E** | 520 | 3.35% | 21 | 0.82% | <0.001 |
| | **CRT2nd Total** | 1625 | | 1028 | | |
| | **CRT2nd LP&E** | 9 | 0.52% | 0 | 0.00% | 0.202 |
| **Follow-up Sample** | **LP&E Samples with a follow-up Test** | 449 | | 36 | | |
| | **Final Diagnosis Positive** | 27 | 6.03% | 4 | 11.11% | 0.306 |
| | **Final Diagnosis Negative***| 386 | 86.16% | 25 | 69.44% | 0.014 |
| | **Final Diagnosis Indeterminate** | 36 | 8.04% | 7 | 19.44% | 0.031 |

**HIV-S**: HIV Screening Serology Results, **CRT:** Creatinine Testing, **LP&E**: Low Positive and Equivocal Results, **HIV-S Only Total:** Total Number of HIV Screening Serology Results without Concurrent Creatinine Testing.

**HIV-S Only LP&E:** Total Number of Low Positive and Equivocal HIV Screening Serology Results without Concurrent Creatinine Testing **HIV-S/CRT total**: Total Number of HIV Screening Serology Result with Concurrent Creatinine Testing, **HIV-S/CRT LP&E:** Low Positive and Equivocal Results Generated by HIV Screening Serology with Concurrent Creatinine Testing **CRT1st Total:** Total number of HIV Screening Serology Results with Concurrent Creatinine Result Released Before HIV Result, **CRT1st LP&E:** Number of Low Positive and Equivocal HIV Screening Serology Results with Concurrent Creatinine Result Released Before HIV Result **CRT2nd Total:** Total number of HIV Screening Serology Results with Concurrent Creatinine Testing with Creatinine Result Released after HIV Result, **CRT2nd LP&E:** Number of Low Positive and Equivocal HIV Screening Serology Results with Concurrent Creatinine Testing with Creatinine Result Released after HIV Result.

*False positive samples.

submitted a median of 5 [IQR 2–22] and maximum of 497 days after the initial sample. Over the study period, which included a series of quality improvement interventions (as mentioned previously), there was a declining trend in the prevalence of LP&E samples. The dedicated sample intervention brought about the most significant change in this rate (Fig 1). Similarly, the prevalence of FP results declined over the study period with the lowest levels reached after the dedicated sample intervention (Fig 2). Moreover, based on measured rates in low positive samples we estimate that before the intervention, of 44 117 HIV screening serology samples, 753 (1.71%) were false positive, declining to 48 of 7 072 samples (0.68%) post-intervention (p<0.01). Of the 235 FP results with concurrent CRT testing, 227 (96%) were CRT1st samples. The overall true positive proportion of the results, reflecting the positive prevalence in the tested population, was 17.9% (9228 of 51189).

Before the dedicated sample intervention, the odds ratio (OR) for obtaining a LP&E HIV-S result vs a negative result was 2.40 (95% CI: 2.09–2.75) in samples with concurrent CRT. Furthermore, the OR for producing FP vs. negative results in samples exposed to concurrent CRT was 2.20 (95% CI: 1.79–2.69). Following the dedicated sample intervention, these odds ratios declined to 0.48 (95% CI: 0.26–0.85) for LP&E and 0.89 (95%CI: 0.41–2.0) for FP results, respectively. This association of LP&E HIV-S results with concurrent CRT decreased significantly as a result of the dedicated sample intervention (p<0.001). In addition, before the

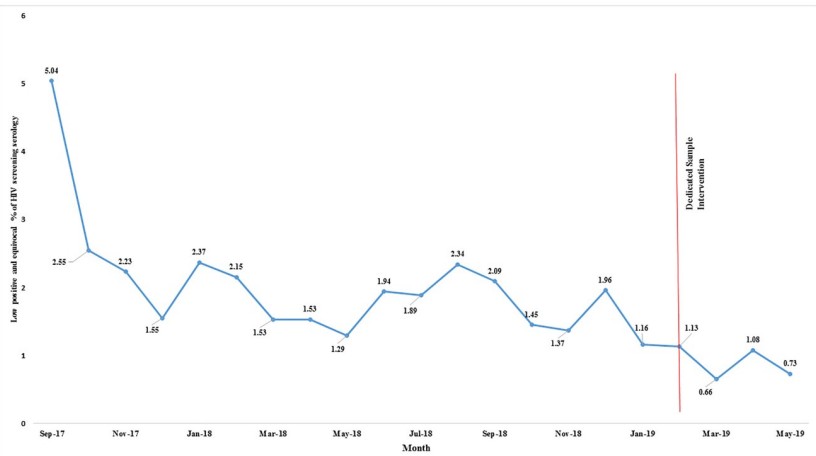

**Fig 1. Low positive and equivocal results monthly proportions displayed over the 21 month study period.** A downward trend in percentage of LP&E results was observed with serial introduction of measures to reduce sample contamination on the analyser.

dedicated sample intervention in samples with CRT1st vs CRT2nd, the OR was 7.04 (95% CI: 3.33–14.87) for LP&E HIV-S results and 4.1 (95% CI: 1.69–9.97) for FP HIV-S result. Taken together this provides strong support that this intervention reduced or prevented contamination.

The manufacturer's assay estimated PPV using this study's seroprevalence is 99.1% whereas the actual PPV in our setting was 91.9%. The manufacturer's claimed specificity was 99.81% (4), compared to the actual specificity of 98.1%. Of 411 results shown to be FP (by testing a different sample or by testing on a different platform), 123 (30%) had same sample concurrent confirmatory serology testing on a separate platform. Of these only 52 (42%) had discordant

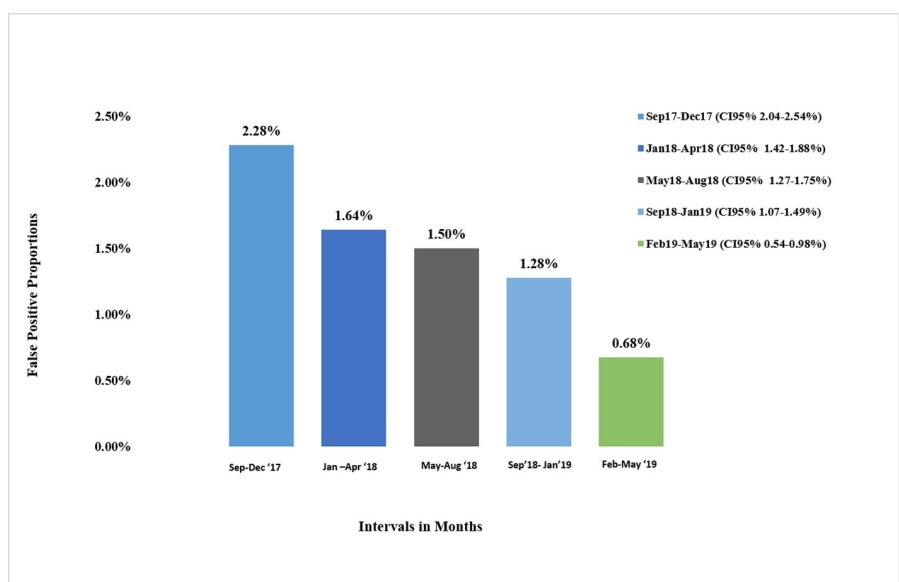

**Fig 2. False positive proportions in groups with the last group being after the dedicated sample intervention.** Bar chart shows the percentage of LP&E results over 4 month intervals from September 2017 to May 2019.

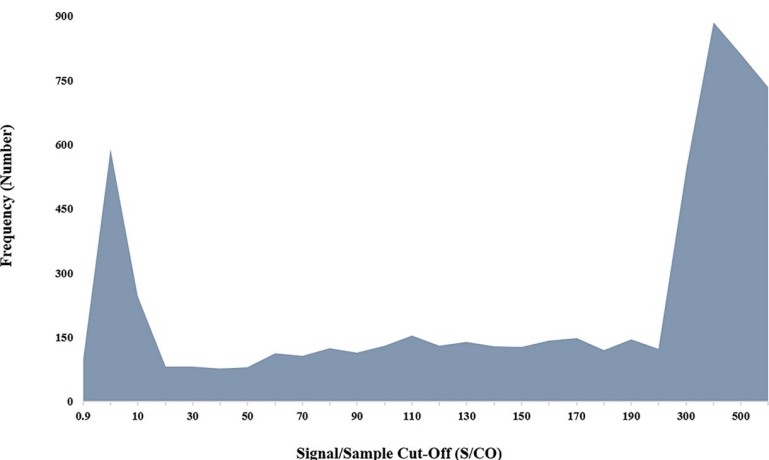

**Fig 3. The frequency of HIV-S results' signal distribution over the positive range.** The chart depicts the bimodal distribution of sample signals peaking in the low positive and equivocal as well as the high positive range.

results indicating that confirmatory testing on the same sample would not identify the majority of false positives.

Positive sample signals showed a bimodal distribution as indicated in Fig 3. The majority of positive results are expected above a S/CO of 200 but the first peak in the low signal range is not expected from an assay with a high reported specificity.

## Discussion

The demand on clinical laboratories to produce hundreds to thousands of results every hour has led to the utilisation of automated high throughput analysers. As evidenced in our study, this can increase the risk of sample cross-contamination. A sample had a significant risk of producing a FP result if previously exposed to the clinical chemistry (CRT) analyser (OR of 2.20 95% CI: 1.79–2.69). The most significant association pointing to sample contamination was the increased risk seen in samples with CRT performed before HIV-S. Samples had an OR of 7.04 (95% CI: 3.33–14.87) for testing LP&E vs negative when comparing CRT1st vs. CRT2nd. If CRT was tested second it could not have been contaminated by the clinical chemistry analyser before being tested for HIV. The success of the "dedicated sample implementation" intervention is demonstrated by the subsequent reduction of LP&E results by 55% and FP results by 60%. A significant reduction of 82% in the LP&E results with concurrent CRT testing after the dedicated sample intervention was observed. Of the LP&E samples that had subsequent tests, 85% were negative, supporting that the large majority of LP&E results are FP. Moreover, same sample confirmatory testing on a different platform averted only 42% of false positives.

The high FP rate, especially before the interventions, leading to a much lower PPV than predicted by the manufacturer's specificity claim, are concerning. Only 51% of clinicians responded by providing a follow-up sample in response to the comment issued with LP&E results; even though most of these responded early (after a median of 5 days), most FP cases were not rectified.

FP results usually generate LP&E S/CO signals as seen by other centres [2, 6, 7]. Using a combination of independent sample testing and different platform testing, for confirmation, we found that 85% of LP&E results were FP and therefore, LP&E result prevalence can be used as an indicator of FP rate. In other settings, the PPV for this assay approached 100% when

using a higher positive threshold than recommended by the manufacturer (S/CO over 30) in European and Asian populations [7–9]. Given the high proportion of FP results among LP&E (S/CO <30) in our setting, the use of a higher threshold for positivity helped to identify cases likely to be false positive in our context, whereas the high proportion that tested negative when testing an independent sample highlights the importance of confirming HIV serology on a separately collected sample. Blaich et al (2017) and Uetwille-Geiger et al (2018) both found a lower than expected specificity of this assay [8, 10]. Patient factors were the major factor associated with FP results in their research. However, in our context instrument sample contamination was the primary cause.

A large sample size (n = 51189) over an extended period of 21 months allowed us to estimate rates and odds ratios with high precision. However, extrapolation of the FP proportion of samples with no subsequent independently collected follow-up testing was limited by uncertainty, minimal patient information and the possibility that patients may be elite controllers or on antiretroviral therapy. Clinicians may have relied on rapid point of care tests at clinics to repeat HIV testing on patients with LP&E results instead of laboratory testing. However this approach is not without risk as the performance of HIV rapid immunochromatographic tests is operator dependent [11, 12]. Rarely, FP results have a high S/CO [13]. However, our focus was to reduce on-platform contamination, which usually results in low S/CO values.

An HIV serology screening assay with such a low PPV is likely to result in an inordinately high number of HIV uninfected patients being diagnosed as positive. Although subsequent testing of an independent sample would reduce this risk, a contamination rate of around 2% at the study start would have resulted in some patients randomly testing positive on the subsequent independently collected follow-up sample and therefore inappropriate 'confirmed diagnosis' and initiation of antiretroviral therapy in HIV uninfected individuals may have happened. The accuracy of HIV serological diagnosis was improved through implementation of multiple measures at TBH NHLS, which resulted in a gradual decline in FP and LP&E prevalence. It may however not be feasible to implement all these measures in all high burden HIV testing settings. The national and international HIV testing guidelines [1, 14] recommend same sample confirmatory testing on a different platform, and confirmation of reactivity on a follow-up sample. However, less than half of the FP results produced discordant confirmation results when the original sample was tested on a separate confirmatory assay as per guidelines. Therefore, if sample contamination occurs, confirmatory testing of the same sample has limited value. Our quality improvement intervention highlights the importance of a dedicated sample for HIV testing, which reduces the risk of contamination and which should be emphasised in national and international HIV programs.

In summary to reduce the risk of contamination and FP results we employed the following strategies:

1. **Setting-appropriate and conservative cut-offs:** The manufacturer regards a signal/sample cut-off (S/CO) > 1 as positive (4). However, based on local verification and data from other sites, a S/CO from 0.9–3) is regarded as equivocal (E), 3–30 as low positive (LP) and ≥ 30 as positive (P) [6].

2. **Dedicated sample requirement for HIV testing:** We issued a notice to clinicians requiring them to send separate blood samples when HIV testing is requested, to reduce the risk of contamination on pre-analytical instrumentation and high throughput chemistry analysers [15].

3. **Training of staff** to ensure that samples for HIV serology are processed only on the dedicated Virology module.

4. **Dedicated sample labelling:** To enforce dedicated sample testing for HIV serology, a suffix (referred to hereafter as "Q" suffix) is added to the barcode label on all dedicated HIV serology samples [16]. This suffix ensures that samples for HIV serology can only be processed on a dedicated virology module. Other modules (clinical chemistry, haematology, immunology, etc.) in the laboratory are unable to process a sample with a suffix on its label; isolating samples from suspected areas of cross-contamination risk.

5. **After requests for HIV serology are no longer permitted on samples already processed on the shared interdisciplinary module.**

The fact that nearly 800 people included in this study interval had FP results and more than 500 people may still not know their true HIV status as there was no subsequent testing, despite having contact with health care, is a concern and action should be taken to prevent this.

## Conclusions

This investigation highlights the risk of using shared platforms, which may be prone to sample contamination affecting HIV serological testing, and emphasizes the need for proper evaluation, environment adapted thresholds for positive vs. indeterminate (LP&E) serology, the importance of expert oversight and quality improvement to ensure that HIV serology is accurate. The implementation of a dedicated sample for HIV serology testing ultimately improved the overall validity of the test results. Further research to understand the properties and mechanisms of instrument-sample contamination to improve platform design and workflow may facilitate the prevention of false positive HIV serology. These instruments are also used to test other viral infections such as Hepatitis B. As shown in our study with regards to HIV, it is conceivable that micro-contamination between positive and negative Hepatitis B surface antigen samples could occur. This too warrants further investigation.

## Supporting information

**S1 Dataset. Dataset of all test results and corresponding values over the study interval.** (XLSX)

## Author Contributions

**Conceptualization:** Michael A. Linström, Wolfgang Preiser, Helena W. Vreede, Stephen N. J. Korsman, Annalise E. Zemlin, Gert U. van Zyl.

**Data curation:** Michael A. Linström.

**Formal analysis:** Michael A. Linström, Gert U. van Zyl.

**Funding acquisition:** Michael A. Linström, Gert U. van Zyl.

**Investigation:** Michael A. Linström.

**Methodology:** Michael A. Linström, Wolfgang Preiser, Stephen N. J. Korsman, Gert U. van Zyl.

**Project administration:** Michael A. Linström.

**Resources:** Michael A. Linström.

**Software:** Michael A. Linström.

**Supervision:** Michael A. Linström, Wolfgang Preiser, Gert U. van Zyl.

**Validation:** Michael A. Linström, Wolfgang Preiser, Gert U. van Zyl.

**Visualization:** Michael A. Linström, Gert U. van Zyl.

**Writing – original draft:** Michael A. Linström.

**Writing – review & editing:** Michael A. Linström, Wolfgang Preiser, Nokwazi N. Nkosi, Helena W. Vreede, Stephen N. J. Korsman, Annalise E. Zemlin, Gert U. van Zyl.

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
