## [Decision Letter · Decision Letter 0]

5 Nov 2020

PONE-D-20-03947

HIV false positive screening serology due to sample contamination reduced by a dedicated sample and platform in a high prevalence environment

PLOS ONE

Dear Dr. Linström,

Thank you for submitting your manuscript to PLOS ONE. After careful consideration, we feel that it has merit but does not fully meet PLOS ONE’s publication criteria as it currently stands. Therefore, we invite you to submit a revised version of the manuscript that addresses the points raised during the review process.

In accordance with reviewer's comments, this manuscript is really in need of clarification (the aim and the style). To that end an extensive english editing is required to render the manuscript readable by non-experts. In addition  sample contaminations are a real diagnostic issue and should be solved using simple means, but which ones? This is not clear as 'proposed' in the conclusion section  < Automated high throughput shared diagnostic platforms risk generating false-positive HIV test results, due to sample contamination. "Measures" are required to address this. Combining separate samples for HIV serology with sample labels restricting testing to dedicated modules successfully reduced false positive HIV rates>. The authors should list what should be the measures, please specify in detail the measures to avoid contamination.

We look forward to receiving your revised manuscript.

Kind regards,

Jean-Luc EPH Darlix, MG, Ph.D.

Academic Editor

PLOS ONE

Journal Requirements:

2. In the ethics statement in the manuscript and in the online submission form, please provide additional information about the patient records/samples used in your retrospective study.

Specifically, please ensure that you have discussed whether all data/samples were fully anonymized before you accessed them and/or whether the IRB or ethics committee waived the requirement for informed consent.

If patients provided informed written consent to have data/samples from their medical records used in research, please include this information.

Reviewers' comments:

Reviewer's Responses to Questions

**Comments to the Author**

1. Is the manuscript technically sound, and do the data support the conclusions?

Reviewer #1: Yes

2. Has the statistical analysis been performed appropriately and rigorously? 

Reviewer #1: Yes

3. Have the authors made all data underlying the findings in their manuscript fully available?

Reviewer #1: Yes

4. Is the manuscript presented in an intelligible fashion and written in standard English?

Reviewer #1: Yes

5. Review Comments to the Author

Reviewer #1: This paper provides good evidence for how false positive HIV serology results can be generated on automated analysers that perform a combination of chemistry, immunology and serology tests frequently used in clinical pathology laboratories. The key finding was the 4-fold increase in likelihood that a sample would test HIV positive if it had first passed through the chemistry module of the analyser (HIV serology tested after creatinine was tested). The authors also demonstrate the effectiveness of measures to reduce sample contamination, in particular use of a dedicated sample tube for HIV serology testing. Recognition of this problem is very important, especially where these instruments are used in areas of high HIV sero-prevalence. I recommend publication, but there are a number of aspects of the manuscript that require fixing. I found the write up of the results particularly difficult to follow

Abstract:

Line 24: Alternative lead in sentence: Automated testing of HIV serology on clinical chemistry analysers has become common.

Line 24: “high throughput, high HIV….”

Line 38: “repeat independent testing on follow up samples of 485…”

Lines 40-42: “The odds ratio for generating a false positive HIV serology result was 4.1 (95%CI:1.69-9.97) when creatinine was tested before HIV serology on the same sample…”

Introduction:

P3, line 51; full stop after “serological testing”

P3, line 80: delete “in order…recently”. Replace with “We issued a notice…”

P3 line 86: Dedicated sample labelling: “ to enforce dedicated sample testing for HIV serology a suffix (referred to hereafter as “Q”suffix) is added to the barcode sample label on all…”

I would prefer the authors to refer to the use of a dedicated sample tube, rather than “Q suffix” (which was the technical means by which this end was achieved) throughout the manuscript, but I don’t feel too strongly about it.

Remove references 11 and 12 – these are not documents available to journal readers.

Line 92: “After requests for HIV serology are not permitted on the…”

P4:

Study aims: (switch around)

1) To assess factors associated with LP&E serology

2) To investigate the impact of measures to ensure the use of a dedicated sample on the overall…”

Methods:

Under data retrieval and analysis:

After the first lead-in sentence, switch the order of paragraphs as follows

section from line 136 to 152 should come before lines 121-135.

Line 128: “…would not have been at risk of contamination from this source”

Line 140 : replace “these samples” with “initial results” and “negative” with “false positive”

Line 142: “the sample was classified as true positive”

Line 145: “It is expected that not all LP&E results have…”

Results:

158: “…before and after the dedicated sample tube intervention…”

162-175: this section is very hard to follow. Please rewrite focussing on the main findings you want to highlight.

I am confused by some of these results. In table 1 and in the text it is stated that a total of 3591 samples were concurrently tested for creatinine “after “Q suffix” implementation (in 2563 creatinine was tested first and in 1028 it was tested second). How can this be? The Q suffix was supposed to ensure that the sample was only tested for HIV. These numbers should all be “0”.

It seems to me that the important figures to compare are the rates of LP&E between the samples where creatinine was tested first, vs where it was tested 2nd . i.e. 3.32% vs 0.52%

Lines 182-191: again very difficult to follow.

Table 1

Title should read: “Summary of low positive and equivocal HIV antibody rates in samples tested before and after introduction of a dedicated sample tube for HIV serology testing.

The figures (numbers) for LP&E results: please amend or make it clear what the numbers in columns 3 and 4 (rows 7-12) refer to.

Middle section of the table:

Column 1 row one: LP&E samples with a repeat test (or positive samples with a repeat test)

Bottom section: (extrapolation data) adds no value. I would remove

Figure 1:

No legend is given. I would suggest “ A downward trend in percentage of LP&E samples was observed with serial introduction of measures to reduce sample contamination on the analyser.”

Figure 2:

Legend: Bar chart shows percentage of LP&E results over 4 month intervals from September 2017 to May 2019.

Discussion:

Rewrite the first paragraph with a lead in sentence contextualising the problem. Highlight the main findings. Don’t restate all the results in the first paragraph.

References:

The following references are not available to journal readers. Either provide a URL or remove from the list: 7, 8, 10, 11, 12,

Provide URL for 4 and 20

6. PLOS authors have the option to publish the peer review history of their article (what does this mean?). If published, this will include your full peer review and any attached files.

Reviewer #1: **Yes: **Diana Hardie

---

## [Author Response · Author response to Decision Letter 0]

7 Dec 2020

Thank you for considering our manuscript and taking time to review it. Herewith, find our response to the reviewer:

“This paper provides good evidence for how false positive HIV serology results can be generated on automated analysers that perform a combination of chemistry, immunology and serology tests frequently used in clinical pathology laboratories. The key finding was the 4-fold increase in likelihood that a sample would test HIV positive if it had first passed through the chemistry module of the analyser (HIV serology tested after creatinine was tested). The authors also demonstrate the effectiveness of measures to reduce sample contamination, in particular use of a dedicated sample tube for HIV serology testing. Recognition of this problem is very important, especially where these instruments are used in areas of high HIV sero-prevalence. I recommend publication, but there are a number of aspects of the manuscript that require fixing. I found the write up of the results particularly difficult to follow”

• We clarified many aspects of the results write up.

‘Abstract:

Line 24: Alternative lead in sentence: Automated testing of HIV serology on clinical chemistry analysers has become common.

Line 24: “high throughput, high HIV….”

Line 38: “repeat independent testing on follow up samples of 485…”

Lines 40-42: “The odds ratio for generating a false positive HIV serology result was 4.1 (95%CI:1.69-9.97) when creatinine was tested before HIV serology on the same sample…”’

• These changes have been made.

• “Automated testing of HIV serology on clinical chemistry analysers has become common. High sample throughput, high HIV prevalence and instrument design could all contribute to sample cross-contamination by microscopic droplet carry-over from seropositive samples to seronegative samples resulting in false positive low-reactive results.”

• “Of 485 initially low positive or equivocal samples 411 (85%) tested negative when retested using an independently collected sample.”

• “The odds ratio for a false positive HIV serology result was 4.1 (95% CI: 1.69-9.97) when creatinine level was determined first, versus not, on the same sample, suggesting contamination on the chemistry analyser.”

‘Introduction:

P3, line 51; full stop after “serological testing”

P3, line 80: delete “in order…recently”. Replace with “We issued a notice…”’

• These changes have been made. 

• “Laboratory-based diagnosis of Human Immunodeficiency Virus (HIV) infection in adults relies on 4th generation serological testing.”

• “Dedicated sample requirement for HIV testing: We issued a notice to clinicians requiring them to send separate blood samples when HIV testing is requested, to reduce the risk of contamination on pre-analytical instrumentation and high throughput chemistry analysers[15].”

“P3 line 86: Dedicated sample labelling: “ to enforce dedicated sample testing for HIV serology a suffix (referred to hereafter as “Q”suffix) is added to the barcode sample label on all…”

I would prefer the authors to refer to the use of a dedicated sample tube, rather than “Q suffix” (which was the technical means by which this end was achieved) throughout the manuscript, but I don’t feel too strongly about it.

Remove references 11 and 12 – these are not documents available to journal readers.”

• We concur and have removed reference to “Q” suffix. We have replaced it with reference to a dedicated separate sample or dedicated sample intervention.

‘Line 92: “After requests for HIV serology are not permitted on the…”’

• These changes have been made.

• “After requests for HIV serology are no longer permitted on samples already processed on the shared interdisciplinary module.”

“P4:

Study aims: (switch around)

1) To assess factors associated with LP&E serology

2) To investigate the impact of measures to ensure the use of a dedicated sample on the overall…”’

• These changes have been made.

• “1) To assess the factors that were associated with low positive and equivocal (LP&E) and FP HIV serology testing. 

2) To investigate the impact of the dedicated sample on the overall prevalence of LP&E and FP results.”

“Methods:

Under data retrieval and analysis:

After the first lead-in sentence, switch the order of paragraphs as follows

section from line 136 to 152 should come before lines 121-135.

Line 128: “…would not have been at risk of contamination from this source”

Line 140 : replace “these samples” with “initial results” and “negative” with “false positive”

Line 142: “the sample was classified as true positive”

Line 145: “It is expected that not all LP&E results have…”’

• These changes have been made.

• “As molecular testing is often required in the work-up of discordant serology, the proportion of LP&E results that had a fresh independently collected sample for subsequent serological or molecular HIV test was assessed. The combined outcome of the subsequent independently collected sample’s test results was then classified as negative, positive, and indeterminate. The median time between the original and subsequent fresh independently collected test result was used to indicate clinician response to LP&E results.

When a subsequent independently collected serology test or molecular test gave a negative result, initial results were classified as FP. When a subsequent independently collected follow-up serology result was high positive and confirmed on the confirmatory assay or the molecular test result was positive, the sample was classified as true positive. If subsequent testing produced an LP&E result like the original, with no additional molecular or serology testing results, samples were classified as indeterminate.

The FP rate was defined as the percentage of LP&E samples with a negative definitive result, as clinicians are required to send independently collected follow-up samples if an LP&E result is generated. It is expected that not all LP&E results have testing of subsequent independently collected follow-up samples due to the reliance on clinicians to provide such samples. Therefore, the false positive rate was extrapolated based on the LP&E results with known subsequent independently collected follow sample testing results. This rate was then used to calculate the prevalence of true positives, the true positive predictive value (PPV) and the specificity. The PPV calculated from the prevalence and the manufacturer’s claimed specificity was used as comparison.

The outcome of confirmatory testing was described as overall proportion of FP results that generated a discordant confirmatory test result on a separate platform, bearing in mind that the sample could have been contaminated before confirmatory testing.

Results for HIV serology including HIV-S and HIV confirmatory (HIV-C) testing and subsequent tests linked to the same individual were retrieved from the NHLS TrakCare laboratory information system. As a single sample identifier (ID) is generated when different sample tubes linked to the same request form are registered, the same sample-ID for different tests indicates concurrent testing. However, it does not necessarily indicate testing of the same tube on a different instrument. Therefore, in order to identify samples at risk of contamination, we first used CRT registered on the same sample ID of HIV-S samples as a screen for being at risk of clinical chemistry module processing and contamination. Second, we used samples being tested on the clinical chemistry module first, rather than second to further stratify contamination risk. We could be confident that samples with CRT results released after HIV-S results did not pass through the clinical chemistry module first and would not have been a source of contamination risk. 

LP&E and FP results were stratified into samples with concurrent CRT testing vs. those without, CRT results released first (CRT1st) vs. CRT results released second (CRT2nd), and before and after the “Q” suffix implementation. These comparisons included prevalences and the calculation of odds ratios (OR) to determine the risk of generating a LP&E and FP result within these strata. Fischer’s Exact Test was used to assess whether there were significant associations with LP&E or FP results, respectively before and after dedicated sample intervention, whereas the Mantel-Haenszel test was used to assess whether these associations were independent of the dedicated sample intervention.”

‘Results:

158: “…before and after the dedicated sample tube intervention…”’

• These changes have been made.

• “Table 1 separates these values into totals found before and after the dedicated sample intervention and illustrates the proportional changes seen.”

“162-175: this section is very hard to follow. Please rewrite focussing on the main findings you want to highlight.

• We have made extensive changes to the article and have attempted to more clearly highlight the main aims of the study. 

I am confused by some of these results. In table 1 and in the text it is stated that a total of 3591 samples were concurrently tested for creatinine “after “Q suffix” implementation (in 2563 creatinine was tested first and in 1028 it was tested second). How can this be? The Q suffix was supposed to ensure that the sample was only tested for HIV. These numbers should all be “0”.

• We have clarified that these samples had creatinine testing requested on them, but due to the intervention a dedicated separate sample was processed. These results illustrate, despite concurrent creatinine testing being requested, there was a significant decline in the number of false positive results after the intervention.

“It seems to me that the important figures to compare are the rates of LP&E between the samples where creatinine was tested first, vs where it was tested 2nd . i.e. 3.32% vs 0.52%”

• We agree and have emphasised these findings.

‘Lines 182-191: again very difficult to follow”’

• We have made significant changes to the text here.

“Table 1

Title should read: “Summary of low positive and equivocal HIV antibody rates in samples tested before and after introduction of a dedicated sample tube for HIV serology testing.

The figures (numbers) for LP&E results: please amend or make it clear what the numbers in columns 3 and 4 (rows 7-12) refer to.

Middle section of the table:

Column 1 row one: LP&E samples with a repeat test (or positive samples with a repeat test)

Bottom section: (extrapolation data) adds no value. I would remove

Figure 1:

No legend is given. I would suggest “ A downward trend in percentage of LP&E samples was observed with serial introduction of measures to reduce sample contamination on the analyser.”

Figure 2:

Legend: Bar chart shows percentage of LP&E results over 4 month intervals from September 2017 to May 2019.”

• Suggestions have been accepted and necessary changes to the table and text have been made.

• “Table 1. Summary of low positive and equivocal (LP&E) HIV antibody rates in samples tested before and after introduction of a dedicated sample tube for HIV serology.

HIV-S: HIV Screening Serology Results, CRT: Creatinine Testing, LP&E: Low Positive and Equivocal Results, HIV-S Only Total: Total Number of HIV Screening Serology Results without Concurrent Creatinine Testing 

HIV-S Only LP&E: Total Number of Low Positive and Equivocal HIV Screening Serology Results without Concurrent Creatinine Testing HIV-S/CRT total: Total Number of HIV Screening Serology Result with Concurrent Creatinine Testing, HIV-S/CRT LP&E: Low Positive and Equivocal Results Generated by HIV Screening Serology with Concurrent Creatinine Testing CRT1st Total: Total number of HIV Screening Serology Results with Concurrent Creatinine Result Released Before HIV Result, CRT1st LP&E: Number of Low Positive and Equivocal HIV Screening Serology Results with Concurrent Creatinine Result Released Before HIV Result CRT2nd Total: Total number of HIV Screening Serology Results with Concurrent Creatinine Testing with Creatinine Result Released after HIV Result, CRT2nd LP&E: Number of Low Positive and Equivocal HIV Screening Serology Results with Concurrent Creatinine Testing with Creatinine Result Released after HIV Result

*False positive samples”

• “Fig1. Low positive and equivocal results monthly proportions displayed over the 21 month study period. A downward trend in percentage of LP&E results was observed with serial introduction of measures to reduce sample contamination on the analyser.”

• “Fig2. False positive proportions in groups with the last group being after the dedicated sample intervention. Bar chart shows the percentage of LP&E results over 4 month intervals from September 2017 to May 2019.

“Discussion:

Rewrite the first paragraph with a lead in sentence contextualising the problem. Highlight the main findings. Don’t restate all the results in the first paragraph.”

• We have rewritten the first paragraph and improved the flow of the text.

• “The demand on clinical laboratories to produce hundreds to thousands of results every hour has led to the utilisation of automated high throughput analysers. As evidenced in our study, this can increase the risk of sample cross-contamination. A sample had a significant risk of producing a FP result if previously exposed to the clinical chemistry (CRT) analyser (OR of 2.20 95% CI: 1.79-2.69). The most significant association pointing to sample contamination was the increased risk seen in samples with CRT performed before HIV-S. Samples had an OR of 7.04 (95% CI: 3.33-14.87) for testing LP&E vs negative when comparing CRT1st vs. CRT2nd. If CRT was tested second it could not have been contaminated by the clinical chemistry analyser before being tested for HIV. The success of the dedicated sample intervention is demonstrated by the subsequent reduction of LP&E results by 55% and FP results by 60%. A significant reduction of 82% in the LP&E results with concurrent CRT testing after the dedicated sample intervention was observed. Of the LP&E samples that had subsequent tests, 85% were negative, supporting that the large majority of LP&E results are FP. Moreover, same sample confirmatory testing on a different platform averted only 42% of false positives.”

• 

“References:

The following references are not available to journal readers. Either provide a URL or remove from the list: 7, 8, 10, 11, 12,

Provide URL for 4 and 20”

• Changes were made.

---

## [Editor Report · Decision Letter 1]

28 Dec 2020

HIV false positive screening serology due to sample contamination reduced by a dedicated sample and platform in a high prevalence environment

PONE-D-20-03947R1

Dear Dr. Linström,

We’re pleased to inform you that your manuscript has been judged scientifically suitable for publication and will be formally accepted for publication once it meets all outstanding technical requirements.

Kind regards,

Jean-Luc EPH Darlix, MG, Ph.D.

Academic Editor

PLOS ONE
---

## [Editor Report · Acceptance letter]

2 Jan 2021

PONE-D-20-03947R1 

HIV false positive screening serology due to sample contamination reduced by a dedicated sample and platform in a high prevalence environment 

Dear Dr. Linström:

I'm pleased to inform you that your manuscript has been deemed suitable for publication in PLOS ONE. Congratulations! Your manuscript is now with our production department. 

Kind regards, 

on behalf of

Professor Jean-Luc EPH Darlix 

Academic Editor

PLOS ONE